# Enhancement of Polypeptide Yield Derived from Rapeseed Meal with Low-Intensity Alternating Magnetic Field

**DOI:** 10.3390/foods11192952

**Published:** 2022-09-21

**Authors:** Lina Guo, Yiting Guo, Ping Wu, Shiyi Liu, Chen Gu, Mengdi Wu, Haile Ma, Ronghai He

**Affiliations:** 1School of Food and Biological Engineering, Jiangsu University, No. 301 Xuefu Road, Zhenjiang 212013, China; 2Institute of Food Physical Processing, Jiangsu University, No. 301 Xuefu Road, Zhenjiang 212013, China; 3School of Food Science and Technology, Jiangsu Agri-animal Husbandry Vocational College, Taizhou 225300, China

**Keywords:** alternating magnetic field, *Bacillus velezensis*, rapeseed meal, solid-state fermentation, polypeptides

## Abstract

The application of physical processing technologies in fermentation is an effective way to improve the quality of substrates. The purpose of the study was to evaluate the feasibility of enhancing the polypeptides of rapeseed meal (RSM) by a low-intensity alternating magnetic field (LF-MF)-assisted solid-state fermentation. A protease-producing strain B16 from RSM was isolated and identified as *Bacillus velezensis* by analyzing its morphology and 16S rDNA sequencing. Then, it was employed in solid-state fermentation for polypeptide production. The results showed that the neutral protease activity could reach 147.48 U/mL when *B.*
*velezensis* was cultured under suitable conditions. The protease activity increased rapidly on the 2.5th day of traditional fermentation, while the polypeptide yield reached the maximum on the third day. The highest polypeptides content was achieved by LF-MF-assisted fermentation at magnetic field intensity 140 Gs, treatment 4 h, magnetic field intervention after 16 h of inoculation, and rotation speed 50 rpm/min, which increased by 18.98% compared with traditional fermentation. Therefore, LF-MF-assisted fermentation effectively enhanced the polypeptide yield. The results suggested that LF-MF technology would be widely used to produce bioactive components from agro-industrial by-products.

## 1. Introduction

Rapeseed, a plant that belongs to the *Brassica napus* L family, also called oilseed rape or canola, has been domesticated for thousands of years [1]. Rapeseed is known as the third-largest oilseed crop after oil palm and soybean, and it is cultivated mainly in China, Canada, India, Australia, and Europe [1]. Rapeseed plays a multifaceted role in the circular economy, providing a source of healthy edible oils, biodiesel, and industrial oils for human consumption [2]. After oil extraction from seeds, the by-product (rapeseed meal, RSM) is employed as a nutritionally rich source of high levels of protein, carbohydrates, minerals, and well-balanced amino acids [3]. Traditionally, it is used mainly as a source of dietary protein in the animal feed in ruminant, poultry, and aquaculture diets because of the existence of antinutrient factors, which essentially restrict its nutrition availability [4]. Meanwhile, alongside the continuous population growth, the demand for dietary protein is on the rise dramatically. Hence, it is projected that RSM is expected to shift from an animal feed resource to an ingredient in many food products to meet future protein requirements [5]. In addition, rapeseed polypeptides are reported to possess various bioactivities, such as antihypertensive activity [6], antioxidant activity [7], and antimicrobial activity [8].

In recent years, an increasing number of qualitative research papers have investigated RSM as a suitable source of high-value edible proteins and bioactive peptides by chemical methods (organic solvents extraction), physical methods (mechanical extrusion, roasting, and water boiling), and enzymatic hydrolysis. However, these processing strategies have some drawbacks, such as destroying protein nutrition, high operating costs, low utilization, and even causing environmental pollution [9]. Solid-state fermentation (SSF) is a promising economic alternative to improve the nutritional and functional properties of RSM and simultaneously reduce undesired antinutrient factors [10]. Unlike submerged fermentation, SSF is a process in which microorganisms grow in an environment with low free water content. It is less susceptible to bacterial contamination and reduces wastewater pollution [11].

Physical processing technologies, such as magnetic field and ultrasound technology, are introduced into traditional fermentation to accelerate the fermentation process and further improve the nutritional and functional properties of substrates [12,13]. The propagation of ultrasonic requires a liquid medium, so it is suitable for submerged fermentation, while the magnetic field could be used in SSF because it travels well in the air. In addition, as one kind of nonthermal processing technology in the food industry, the magnetic field has a profound penetration ability to materials and has a broad range of applications [14]. Many studies have investigated the potential applications of magnetic fields on the viability and metabolite production in microorganisms [13,14,15,16]. However, the existing magnetic field equipment used in fermentation lacks the rotation process of the magnetic field, and the fundamental role of the magnetic field cannot be exerted well. Furthermore, studies of magnetic fields on rapeseed meals in SSF have rarely been reported.

In this study, a strain with high protease production ability was isolated and identified for fermentative manufacturing polypeptides from RSM. In addition, the optimization of traditional fermentation conditions and low-intensity alternating magnetic field (LF-MF)-assisted RSM fermentation parameters (magnetic induction, exposure time, rotated speed, exposure period, and treatment modalities) were studied to produce polypeptides. It is hoped that the results presented in this paper will provide a novel, reliable method for improving bioactive peptides.

## 2. Materials and Methods

### 2.1. Materials and Reagents

Defatted rapeseed meal was purchased from Dujiang Albumen Powder Factory (Nanjing, China). Culture mediums were obtained from Sigma Chemical Co. (St. Louis, MO, USA). The chemicals used in this study were all analytical grade and were purchased from Sinopharm Chemical Reagent Co. (Shanghai, China).

### 2.2. Selection of Bacterial Strain

#### 2.2.1. Selection and isolation of strains with high proteinase

Ten grams of RSM were dissolved in 100 mL of LB liquid medium (1% tryptone, 0.5% yeast extract, and 1% NaCl) and shake-flask cultivated at 37 °C, 180 rpm for 1 h. Afterward, the solution was diluted (10^−1^–10^−7^) with sterile water. The suitable gradient dilution (100 μL) was spread on the selection medium agar plates at 37 °C for 24 h. Colonies of different forms were selected for streaking cultivation, and the operation was repeated three times to obtain a single colony.

#### 2.2.2. Preliminary Screening of Strains

The single colonies of different morphologies were inoculated in the medium (100 mL) for 24 h. After cultivation, 1 μL of the solution was spotted in six different positions on a plate of casein protein medium (beef extract 3 g/L, casein 8 g/L, agar 15 g/L, pH 7.0–7.2), starch medium (beef extract 5 g/L, soybean peptone 10 g/L, sodium chloride 5 g/L, solution starch 2 g/L, agar 15 g/L, pH 7.0–7.2), and sodium carboxymethyl cellulose medium (soybean peptone 5 g/L, yeast 0.5 g/L, potassium dihydrogen phosphate 0.2 g/L, sodium chloride 5 g/L, sodium carboxymethyl cellulose 10 g/L, agar 15 g/L, pH 7.0–7.2) respectively, cultivating at 37 °C for 24 h. The ratio of hydrolytic zone diameter to colony diameter (H/C) was measured to evaluate the enzyme activity of the strains preliminarily.

#### 2.2.3. Rescreening of Strains

The protease released by microorganisms in LB liquid medium was determined by way of enzymatic hydrolysis of casein to tyrosine according to the method of Hou et al. [17] with some modifications. The fermentation broth incubating for 24 h was centrifuged at 10,000 rpm for 10 min. Subsequently, the supernatant sample was collected to measure the protease activity. An aliquot of 1 mL of casein solution (10 g/L) and 1 mL of fermentation supernatant was mixed and incubated at 40 °C for 10 min, immediately 2 mL of trichloroacetic acid (65.4 g/L) was added to terminate the reaction. After 10 min of rest, the mixture was filtered with slow qualitative filter paper, and the supernatant was reserved for further analysis. An aliquot of 1 mL of supernatant was mixed with 5 mL of sodium carbonate (42.4 g/L) and 1 mL of forint-phenol reagent and incubated at 40 °C for 20 min. The absorbance was measured at 680 nm by spectrophotometry. The control group added trichloroacetic acid first to the casein solution before the addition of the sample. The absorbance results were compared with the tyrosine standard curve. All samples were tested in triplicate. The activity of protease was calculated as the following formula:*E* = (*A* × 4 × *n*)/(*V* × 10)(1)
where *E* was the activity of protease (U/mL), *A* was the activity of the enzyme in the final diluent of the sample obtained from the standard curve of tyrosine (U/mL), *V* was the volume of the sample (mL), the total reaction volume was 4 mL, *n* was the dilution ratio, and the reaction time was 10 min.

### 2.3. Identification of the Selected Strain

#### 2.3.1. Morphology

The strain named B16 from RSM presented the potential production of protease activity. The suitable gradient dilution of the strain was coated on the LB solid medium and cultivated at 37 °C for 24 h. The characteristics of the single colony were recorded, and Gram-staining and malachite green spore staining were used to observe the microscopic morphology of the strain.

#### 2.3.2. 16S rDNA Sequence Analysis

A genome DNA extraction kit was used to extract and purify the genomic DNA by the instructions. The extracted DNA was used as the template for PCR, 16 S rDNA amplified by upstream primer 27F (5′-AGAGTTTGATCCTGGCTCAG-3′) and downstream primer 1492R (5′-GGTTACCTTGTTACGACTT-3′). DNA sequencing was carried out by Sangon Biotechnology Co. (Shanghai, China). The results were analyzed by BLAST in the NCBI.

### 2.4. Optimization of Cultural Conditions

The selected strain was cultured in LB liquid medium and stored in 30% glycerin with a ratio of 1:1 at −20 °C. The strain was grown in 100 mL of LB liquid medium before inoculation. The activated strain was inoculated in 1% (*v*/*v*) of six mediums and cultured at 37 °C and 180 r/min for 24 h. The total number of colonies, protease activity, and pH were determined to screen the optimal medium. Afterward, the optimal culture conditions of the bacterium were screened, including incubation temperature, incubation dose, initial pH, and shaker speed.

### 2.5. Traditional RSM Solid-State Fermentation

Forty grams of RSM were sterilized at 121 °C for 20 min. The fermentation was conducted in a 250 mL beaker covered with high-temperature resistant tissue sealing films. The method of varying one parameter at a time was adopted to optimize fermentation conditions, including particle size of RSM, fermentation temperature, incubation dose, feed-to-water ratio, and fermentation time. Non-fermented sterilized RSM was used as the control, and each treatment was conducted in triplicate. After the fermentation, RSM was dried and crushed for further analysis.

### 2.6. Low-Intensity Alternating Magnetic-Field-Assisted RSM Fermentation

#### 2.6.1. Low-Intensity Alternating Magnetic Field Set-Up

LF-MF device is an electromagnetic rotating bed coupled with triple coil assemblies manufactured by Jiangsu University. The picture and schematic diagram of the device are shown in Figure 1. The equipment consists of a voltage regulator, a worktable, a magnetic field generation system, a heating system, and a transmission system. The magnetic induction intensity of the magnetic field generation system shows an excellent linear relationship with voltage (R^2^ = 0.9993), which indicates that the corresponding magnetic field intensity is stable at any specific voltage. The transmission system is used to drive the rotation of the material carrier put into the magnetic coil cavity, which makes the material and the Helmholtz coil assembly have a relative cutting effect, enhancing the fermentation processing effect. The heating system is designed to control the temperature inside the magnetic coil cavity in which the fermented samples are treated and provide optimal growth conditions for microorganisms.

#### 2.6.2. Optimization of Magnetic Field Parameters

The magnetic field parameters, including magnetic field intensity, exposure time, rotated speed, and initiation time of magnetic field intervention, were discussed under the optimization results of traditional fermentation. The magnetic field treatment single factor optimization strategy is displayed in Table 1. The fermentation was conducted under the following conditions to optimize the magnetic field intensity: magnetic field intensity in the range from 10 Gs to 160 Gs (fixed parameter: magnetic treatment continuously for 1 h at the magnetic intervention time of 4 h after inoculation; the rotation speed was 50 rpm/min). After 4 h fermentation in a constant temperature incubator, RSM was placed in the triple rotating magnetic field bed (Figure 1) to continue the magnetic-field-assisted fermentation, with exposure time ranging from 1 h to 10 h (Fixed parameter: magnetic field intensity of 140 Gs, the magnetic intervention time of 4 h after inoculation, the rotation speed was 50 rpm/min). Subsequently, the beakers were returned to the original constant temperature incubator to continue fermentation until the end of fermentation. The magnetic rotation speed ranged from 0 to 90 rpm/min, and other parameters were magnetic field intensity of 140 Gs, magnetic field continuously 4 h, and magnetic intervention time of 4 h after inoculation. The initiation time of magnetic field intervention ranged from 0 h to 24 h after inoculation, respectively, with the magnetic field intensity of 140 Gs, magnetic field continuously 4 h, and the rotation speed of 50 rpm/min. The peptide content of RSM after fermentation was determined to optimize the magnetic-field-assisted fermentation parameters.

### 2.7. Determination of Polypeptides Concentration

Two grams of RSM were added into 40 mL distilled water, stirring for 30 min with a magnetic stirrer. Then the mixture was centrifuged at 4000 rpm/min for 10 min. The supernatant and trichloroacetic acid (10%, *w*/*v*) were mixed in an equal volume to denature the protein. After centrifugation (4000 rpm, 10 min), 1 mL supernatant was mixed with reagent A (2% Na_2_CO_3_, 0.1 M NaOH) and incubated in a water bath at 30 °C for 10 min. Then 0.5 mL reagent B (0.5 mL CuSO_4_•5H_2_O, 1% sodium tartrate) was quickly jetted and incubated at 30 °C for 30 min. The absorbance of the solution was determined at 680 nm compared with the standard curve of bovine serum albumin (y = 0.003x + 0.0243, R^2^ = 0.9960).

### 2.8. Statistical Analysis

All experiments were replicated at least three times. Data were presented as the mean ± SD. The results were analyzed using a one-way variance (ANOVA) analysis with Duncan′s new multiple range test by SPSS 13.0 software (IBM, Armonk, NY, USA). All figures were performed with Origin Pro Software Version 8.0 (Origin Lab Corp., MA, USA). A significant level of *p* < 0.05 was considered an indication of statistical significance.

## 3. Results and Discussion

### 3.1. Selection of Strains with High Proteinase Activity

Nineteen strains of bacteria were preliminarily screened from RSM, of which twelve strains were able to produce hydrolytic circles on casein protein medium, starch medium, and sodium carboxymethyl cellulose medium. As shown in Table 2, these 12 strains had potential decomposition ability of casein medium, and the ratios of hydrolytic zone diameter to colony diameter (H/C) were greater than 3.00. In addition, the results indicated that all of them could secret amylase and cellulase to some extent, except for protease. To further clarify the protease production ability of these twelve strains, their enzyme activities were measured, including neutral protease, alkaline protease, and acid protease. It can be seen from Table 3 that these bacteria mainly generated neutral proteases, followed by alkaline proteases. Four strains, named B16, BY4, BY1, and BY3, had significant protease production ability, especially strain B16 showed the highest proteinase activity (25.61 U/mL). Therefore, it was chosen for further study.

### 3.2. Morphology and Identification of Strain

The colony morphology of strain B16 (Figure 2C) was round on LB solid medium with irregular edges, opaque, rough matte, and protruding surface. After Gram staining, the bacteria showed purple (Figure 2D) observed by a microscope, demonstrating a Gram-positive bacterium with a rod-shaped structure. In addition, it was found that there were green zones (Figure 2E) after malachite green staining, indicating that the bacterium produced spores.

The sequence 16S rDNA gene of strain B16 was 1351 bp (Figure 2F). The 16S rDNA sequence was deposited in NCBI GenBank with the accession Bacillus. velezensis (MT649755.1), B. siamensis (MT645306.1), *B. velezensis* (CP054714.1), B. amyloliquefaciens (MT579842.1). Therefore, B16 was designated as *B.*
*velezensis* of Bacillus, based on the morphological and phylogenetic of the 16S rDNA sequence.

### 3.3. Optimization of Cultural Conditions

#### 3.3.1. Culture Medium

Excellent medium and culture conditions can not only enhance the biomass of bacteria but promote the secretion of proteases, which is the premise of fermentation to produce polypeptides [18]. Hence, it was necessary to evaluate the suitable medium and conditions for *B.*
*velezensis*. The enzyme activity, biomass, and pH of *B.*
*velezensis* in different media for 24 h are displayed in Figure 3. The results showed that all of the six media provide nutrition and energy for the growth, multiplication, and metabolism of *B. velezensis*. As can be seen from Figure 3B, biomass cultured in the medium numbered from 3 to 6 was significantly higher than that in medium No. 1 and medium No. 2. In addition, after cultivation in the first five media, the pH of the bacterial supernatant remained above 7, while the pH of the sixth medium decreased to 5.5. The neutral protease and alkaline protease activities in medium No. 6 for 24 h were 74.45 U/mL and 16.04 U/mL, respectively, which were dramatically higher than those in the other five media in terms of enzyme ability (Figure 3A). It can also be found that the yield of neutral protease produced by *B.*
*velezensis* was significantly higher than that of alkaline protease, which might be caused by the weak acidic condition of the medium. In short, medium No. 6 was selected as the suitable medium for *B.*
*velezensis* where the bacteria could produce the highest content of neutral protease and alkaline protease compared with other mediums.

#### 3.3.2. Incubation Temperature

The single-factor test was applied to optimize the culture conditions that affected the protease activity and biomass. *B.*
*velezensis* was grown in YG medium for 24 h, and the results were presented in Figure 4. Temperature is one of the main factors that has a remarkable effect on the growth of microorganisms [19]. As shown in Figure 4A, biomass significantly increased over increased temperature. As for enzyme activity, *B.*
*velezensis* produced the highest content of neutral protease at 34 °C. When the culture temperature was at 28 °C or 40 °C, the biomass was high, but the enzyme activity was low. It might be due to the fact that low incubation temperature caused the poor growth of the bacterium, resulting in a weak secretion of protease. When the temperature was high, the growth and metabolism of the microorganism were inhibited [17]. There was no significant difference in enzyme activity between the culture temperature at 34 °C and 37 °C. Therefore, 34 °C was chosen as the optimum temperature for incubation.

#### 3.3.3. Inoculation Dose and Initial pH in the Medium

The inoculum level is also an essential factor in microbial growth and metabolism. As shown in Figure 4B, the protease activity of the strain enhanced significantly to reach its maximum value (144.80 U/mL) as the inoculation dose increased from 1% to 3%. In addition, there was no increase in the neutral protease activity when the inoculum was further increased to 9%. Hence, the optimal inoculation dose was 5% for the highest neutral protease production by *B. velezensis*.

The pH could influence the ionic state of the medium, structure, morphology, as well as some physiological functions of microorganisms. It also affects the nutrient uptake and product biosynthesis of microorganisms [19]. Five pH levels ranging from 5.0 to 9.0 were investigated in this study. The results in Figure 4C elucidated that when the initial pH values of the medium were 5, 6, or 7, the biomass was significantly higher than that of the medium with pH values of 8 and 9, suggesting the alkaline environment was not suitable for the growth of the strain. In terms of enzyme activity, the figure showed a significant rise from pH 5 to pH 6, and the maximal neutral protease was achieved (144.53 U/mL) at pH 6. Afterward, it was a gradual decline with the increase in the medium pH, which was consistent with that of biomass.

#### 3.3.4. Shaker Speed during Incubation

Shaker speed is another crucial physical factor in the inoculation of bacteria, which plays a vital role in the dissolved oxygen content of the culture medium. Additionally, it could maintain a concentration gradient between the interior and exterior of cells [20]. As shown in Figure 4D, the biomass was above 9 logs CFU/mL at different shaker speeds. The enzyme activity was the lowest at 140 rpm/min because the weak agitation resulted in insufficient dissolved oxygen in the medium, which negatively affected the metabolism. When the shaker speed was at 220 rpm/min, the higher enzyme activity was also not achieved. It was because intense agitation speed enhanced the shear force and dissolved oxygen, which promoted the bacterium into the attenuation stage [19]. Moderate shaker speed could create a good balance between shear force and oxygen transfer. Therefore, the agitation speed at 180 rpm was selected as the optimal shaker speed.

### 3.4. Optimization of the Traditional Fermentation Process

#### 3.4.1. The Particle Size of the Substrate

The primary fermentation conditions were fermentation temperature of 37 °C, inoculum dose of 10 mL/100 g (108 CFU/mL), the ratio of substrate to water 1:1 (40 g/40 mL), and the fermentation time of 2.5 d. The effect of particle size of the substrate on the polypeptide yield is presented in Figure 5A. The results showed that the polypeptide yield significantly decreased with the decrease in particle size of raw material. When the substrate was not sieved, the content of polypeptides after fermentation was 77.94 mg/g, which exhibited no significant difference from the polypeptides yield after the substrate through a 10-mesh sieve. The particle size of the substrate directly influences the utilization of materials by microorganisms. If the substrate particle was too small, the space between the solid particles would be compressed, which could cause a lack of oxygen in the substrate internal and impact the fermentation process. The above results suggested the optimum substrate particle was raw material without sieving.

#### 3.4.2. Fermentation Temperature

It could be found that fermentation temperature played a crucial role in the yield of polypeptides during solid-state fermentation. The suitable temperature was selected by fixing other fermentation conditions. As shown in Figure 5B, polypeptide content markedly increased with the temperature, and it was up to the highest (79.18 mg/g) at 37 °C. Afterward, the yield of polypeptides dramatically decreased to (40.99 mg/g) at 40 °C. It might be due to the fact that higher temperatures accelerated the evaporation of water in the substrate, resulting in a poor fermentation capacity of *B.*
*velezensis*. On the other hand, higher temperature itself was not conducive to the growth and metabolism of microorganisms. In this study, the suitable fermentation temperature was 37 °C.

#### 3.4.3. Biomass Amounts and Substrate/Water Ratio

The effects of various inoculation amount and substrate/water ratios on the production of rapeseed polypeptides by *B. velezensis* were displayed in Figure 5C and Figure 5D. Obviously, the yield of polypeptides increased as the inoculum size increased, with the maximal production when the inoculum concentration was 20%. The density of microorganisms in the fermentation matrix was insufficient to stimulate rapid cell growth and synthesis of extracellular and extracellular enzymes when the inoculum size was less than 20% [21]. The polypeptide content showed a gradually upward tendency with the increase in the water column added to the substrate when the ratio of matrix to water was from 1:0.5 to 1:1.25, and it reached the maximum with the material/water 1:1.25. *B. velezensis* could make full use of water and nutrients in the matrix with sufficient water to have a better effect of fermentation to produce polypeptides.

#### 3.4.4. Fermentation Time

As shown in Figure 5E, the yield of polypeptides climbed up slowly in the first 2 days of fermentation, but it rose sharply on the 2.5 d and reached the maximum on the 3rd day. After continued fermentation, the polypeptide content remained stable until the 4.5th day, when it declined. Figure 6A shows that the neutral protease and alkaline protease activities increased rapidly when the fermentation time was 2.5 d, which promoted the decomposition of fermentation substrates, such as the degradation of proteins to peptides [22] and degradation of trypsin inhibition [23]. The polypeptides content (Figure 6B) decreasing with the prolongation of fermentation was attributed to the fact that polypeptides were efficiently utilized by increasing microorganisms because of their low molecular weight [17]. It was also due to the excessive production of proteases by *B. velezensis*, resulting in the over-hydrolysis of proteins into amino acids. Meanwhile, during the fermentation process, the pH (Figure 6B) in the matrix increased gradually all the time, which indicated that the composition of the substrate changed to some extent.

### 3.5. Effects of LF-MF Assisted Fermentation on Polypeptides

Figure 7A presents the effect of magnetic field intensity on the content of the rapeseed polypeptides in the fermentation process. The primary magnetic-field-assisted fermentation conditions were magnetic field treatment for 1 h, magnetic field intervention after 4 h of inoculation, and the magnetic rotation speed of 50 rpm/min. The results indicated that when the magnetic field intensity was 140 Gs, the polypeptide content was significantly higher than the control group (without magnetic field treatments). Generally speaking, it was apparent that the yield of polypeptides decreased sharply, even lower than that of the traditional fermentation, when the magnetic field intensity was in the range of 40–60 Gs and 110–130 Gs, respectively. This phenomenon could be explained by the “window effect”, which indicated that the target only responded to certain electromagnetic waves of some discrete intensity ranges [24]. These “intensity windows” could affect the expression levels of critical genes in metabolism-related pathways of microorganisms, which had a positive or negative effect on the production of proteases or other main products regulated by them [25].

The effects of different magnetic treatment duration on solid-state fermentation polypeptide production are presented in Figure 7B. There was no significant difference in polypeptide content when the fermentation process was assisted by a magnetic field for 1, 2, and 3 h, respectively, compared with the traditional control group, which might be because the low-intensity magnetic field stimulated the microorganism for a short time. It did not reach the threshold of the biological effect of the magnetic field [14]. After 4 h of magnetic field treatment, the polypeptide content was markedly increased and then decreased gradually with the extension of magnetic treatment duration. It was attributed to the permeability of the cell membrane changes under the magnetic field stress, which promoted the efflux of intracellular proteins [26]. However, the prolonged stimulation inhibited bacterial metabolism, reducing the total intracellular protein content. When the magnetic treatment time was nine hours and ten hours, the polypeptide content increased again, which might be because the microorganism was exposed to the magnetic field environment for a long time, promoting the enhancement of microbial stress ability and adaptability.

A rotating magnetic field generated the eddy current in the exposed substrate and created a magneto-hydrodynamic system in a medium [27]. The rotating speed determined the speed of the cutting magnetic induction line, which could generate an electric current, also called electromagnetic induction. As shown in Figure 7C, when the rotation speed was 50 rpm/min, the polypeptide content reached the maximum, suggesting that a moderate pace could promote the growth of the cells and increase the efficiency of metabolite production. Additionally, it was noted that after 16 h of traditional fermentation, the magnetic field was carried out, and the yield of polypeptides increased significantly. Before 16 h, the microorganism was in the lag phase of the growth process, and the polypeptide content remained stable, and then it was the logarithmic phase of bacterial growth. During this period, the magnetic field treatment of the microorganisms could cause a large amount of protease to secrete, thereby increasing polypeptide production.

### 3.6. Possible Mechanisms

Nowadays, a growing number of scholars and experts pay much attention to suitable magnetic field parameters that can promote microbial growth and metabolism. Canli and Kurbanoglu [15], using a 7 mT static magnetic field to assist SSF of leek with Geotrichum candidum, showed that inulinase enzyme activity increased by about 29.8%. Zhang et al. [28] found that the production of yellow pigment and red pigment increased by 72.5% and 77.3% at the end of the 11th day of fermentation after the low-intensity alternating magnetic field treatment (0.4 mT) on the 8–9th day of SSF by Monascus purpureus. These research studies presented that different magnetic field intensities had various biological effects on the microorganisms. In our study, the magnetic field intensity at 140 Gs could better enhance the yield of rapeseed polypeptides during the solid-state fermentation process. The possible mechanism was that a low-intensity alternation magnetic field could generate positive biological effects on *B.*
*velezensis*. As the first messenger, the extracellular electromagnetic signal was converted into intracellular signal molecules through transmembrane receptor proteins on the cell surface, namely the second messengers (such as Ca^2+^, CAMP), which initiated a series of biological effects cell [29]. The stimulation of the magnetic field might accelerate the transcription and translation process. In addition, it also changed the permeability of the cell membrane, which can secrete the protein more quickly outside the cell, thereby producing a large number of proteases to decompose the substrate into polypeptides. On the one hand, the increase in polypeptide content might be due to the direct effect of the magnetic field on proteases, such as the conformation of enzymes and the influence of van der Waals forces between nonpolar side chains. Alternatively, it could affect the bond of water molecules with the enzyme to maintain the natural conformation of the enzyme molecule, changing the length and strength of the hydrogen bond of the water molecules to change the conformation of the enzyme [25].

## 4. Conclusions

In this study, a strain identified as *B. velezensis* was isolated from RSM by its morphology and 16S rDNA sequencing. Under the ideal growing conditions, the strain exhibited the highest enzyme activity and generated neutral protease primarily when compared to other strains. Additionally, the highest yield of polypeptides was obtained with the low-intensity alternating magnetic-field-assisted fermentation compared with the traditional fermentation. Therefore, LF-MF can be employed as a potential processing technique in the future to produce polypeptides. However, the specific mechanism of promoting the growth of microorganisms and secondary metabolites is not clear yet. Disciplines such as bioinformatics and proteomics will be applied to explore the mechanisms in further research.

## Figures and Tables

**Figure 1 foods-11-02952-f001:**
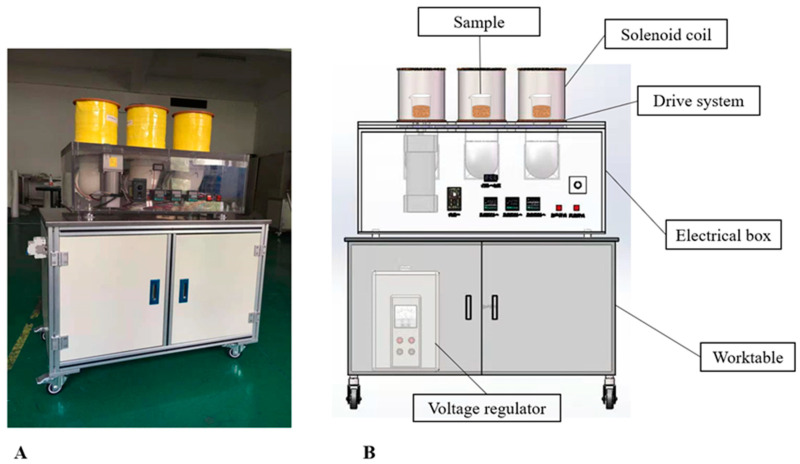
Low intensity rotating magnetic field device. (**A**) The photo of the device; (**B**) the schematic diagram of the device.

**Figure 2 foods-11-02952-f002:**
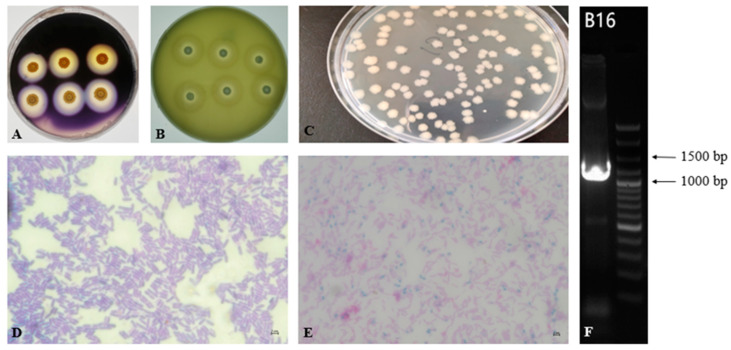
The shapes of strain B16 (Bacillus. velezensis) and the result of the 16S rDNA agarose gel electrophoresis. Hydrolysis circle of amylase (**A**) and protease (**B**) secreted by *B.*
*velezensis*; (**C**): colony morphology of *B.*
*velezensis* observed directly; (**D**,**E**): microscope observation of *B.*
*velezensis* after staining; (**F**): the result of 16S rDNA agarose gel electrophoresis of *B.*
*velezensis*.

**Figure 3 foods-11-02952-f003:**
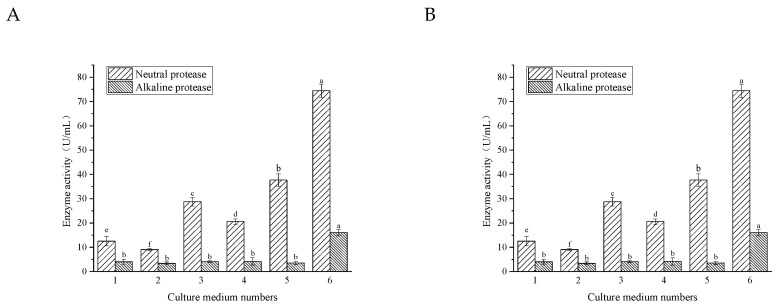
Enzyme activity (**A**), biomass, and pH (**B**) after inoculation with six mediums. The numbers represent different cultural mediums. 1: Beef extract-peptone medium; 2: Nutrient broth medium; 3: LB medium; 4: Typtone soybean medium; 5: Peptone yeast extract glucose medium; 6: YG medium. Bars with different letters differ significantly (*p* < 0.05).

**Figure 4 foods-11-02952-f004:**
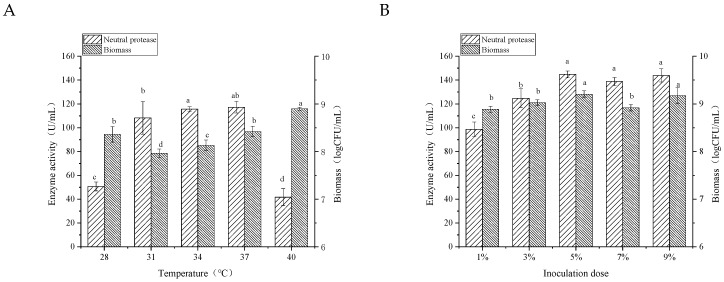
Effects of temperature (**A**), inoculation dose (**B**), initial pH (**C**), and shaker speed (**D**) on the enzyme activity and biomass of *B.*
*velezensis*. Bars with different letters differ significantly (*p* < 0.05).

**Figure 5 foods-11-02952-f005:**
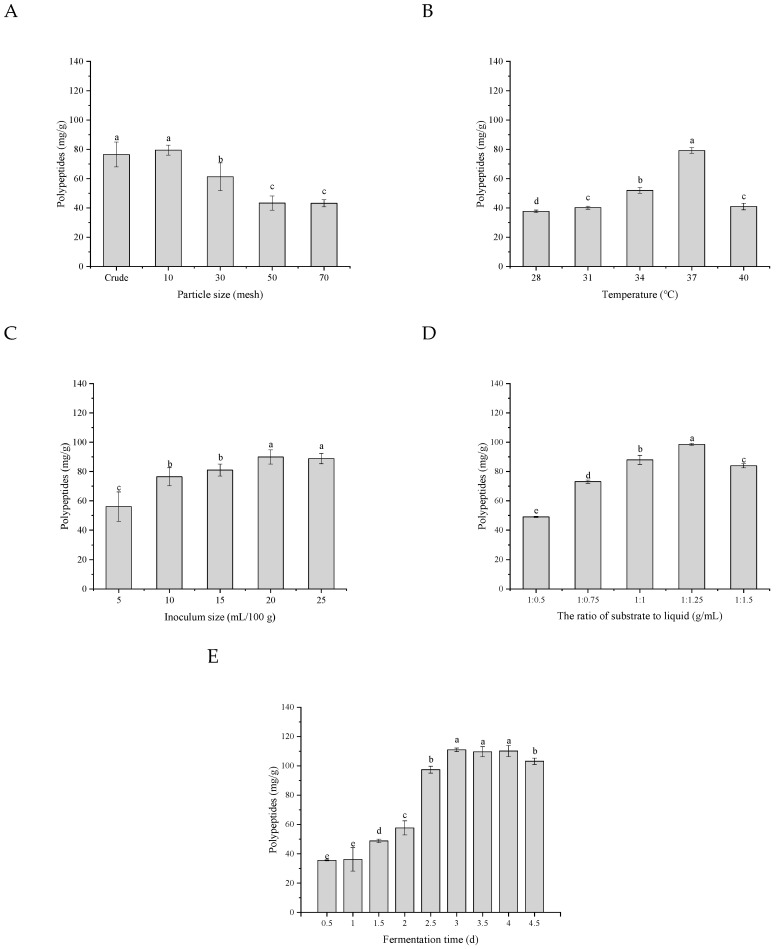
Effects of particle size of the matrix (**A**), fermentation temperature (**B**), inoculum size (**C**), the ratio of substrate to liquid (**D**), and fermentation time (**E**) on polypeptides after SSF by *B.*
*velezensis*. Bars with different letters differ significantly (*p* < 0.05).

**Figure 6 foods-11-02952-f006:**
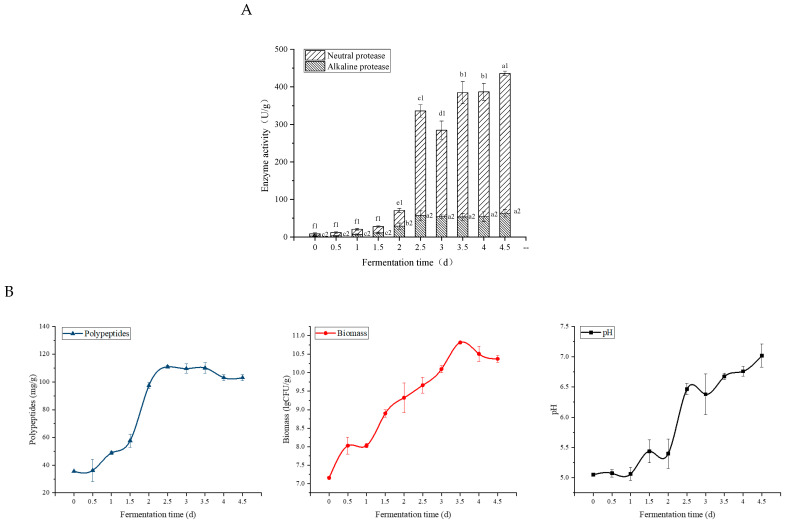
The variation of enzyme activity of fermented RSM (**A**) and the variation of polypeptides, biomass, and pH in fermented RSM (**B**). Bars with different letters differ significantly (*p* < 0.05).

**Figure 7 foods-11-02952-f007:**
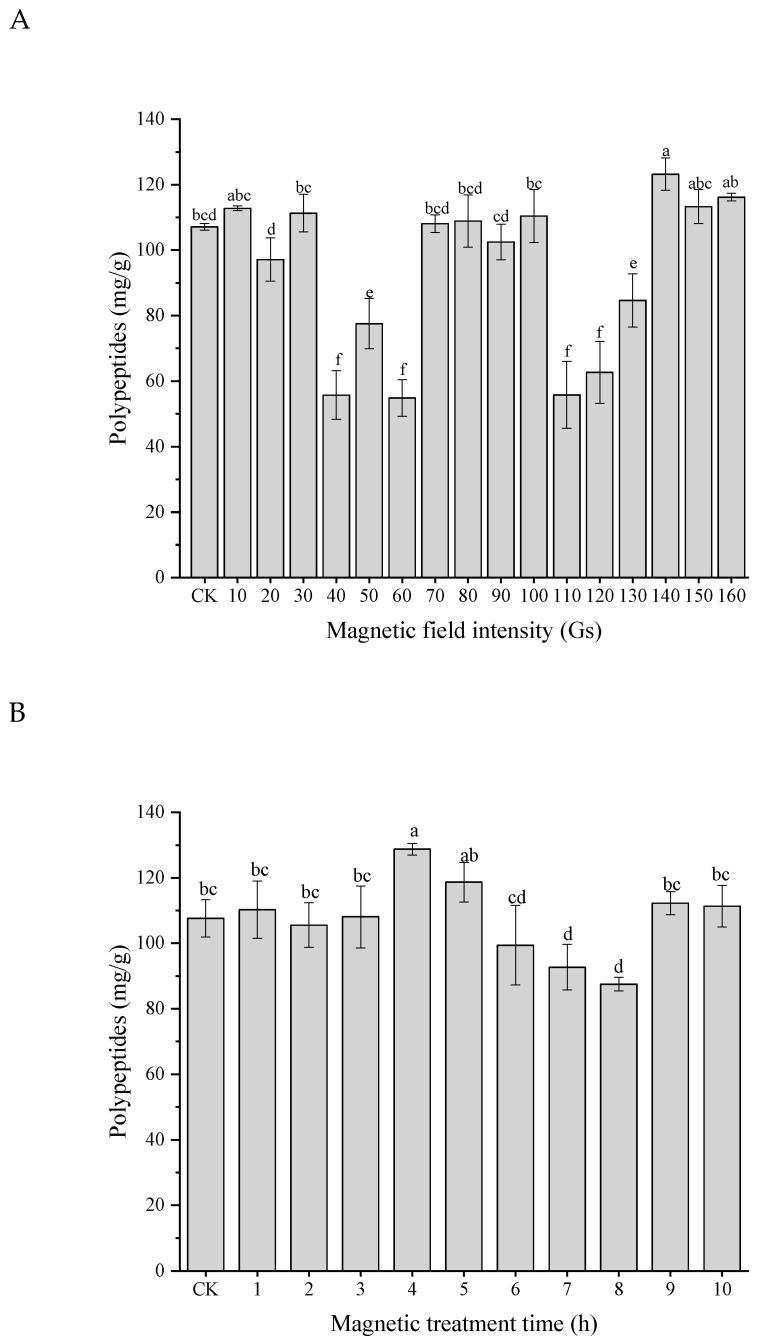
Effects of magnetic field intensity (**A**), duration (**B**), rotation speed (**C**), and initiation time of intervention (**D**) on polypeptide production by LF-MF assisted fermentation. Bars with different letters differ significantly (*p* < 0.05).

**Table 1 foods-11-02952-t001:** The magnetic field treatment single factor optimization strategy.

Single Factor	Treatments	Fixed Parameters
Magnetic field intensity (Gs)	CK, 10, 20, 30, 40, 50, 60, 70, 80, 90, 100,110, 120, 130, 140, 150, 160	LF-MF exposure time: 1 h; Rotation speed: 50 rpm/min; Intervention time: 4 h
LF-MF exposure time (h)	CK, 1, 2, 3, 4, 5, 6, 7, 8, 9, 10	Magnetic field intensity: 140 Gs; Rotation speed: 50 rpm/min; Intervention time: 4 h
Rotation speed (rpm/min)	CK, 0, 10, 30, 50, 70	Magnetic field intensity: 140 Gs; LF-MF exposure time: 4 h; Intervention time: 4 h
Intervention time (h)	CK, 0, 2, 4, 6, 8, 10, 12, 14, 16, 18, 20, 22, 24	Magnetic field intensity: 140 Gs; LF-MF exposure time: 4 h; Rotation speed: 50 rpm/min;

**Table 2 foods-11-02952-t002:** Production capacities of three enzymes secreted by strains (expressed by the ratio of transparent circle diameter to bacterial circle diameter).

Strains	Protease	Amylase	Cellulase
B1	3.25 ± 0.26 cd	2.07 ± 0.20 cd	2.03 ± 0.09 bc
B2	3.26 ± 0.21 cd	2.18 ± 0.18 c	2.00 ± 0.09 bcd
B5	3.57 ± 0.14 bc	2.38 ± 0.13 b	1.88 ± 0.19 d
B9	3.17 ± 0.17 d	1.79 ± 0.10 e	1.93 ± 0.15 cd
B10	3.60 ± 0.40 b	1.96 ± 0.09 de	2.09 ± 0.07 bc
B13	4.00 ± 0.34 a	2.50 ± 0.08 ab	2.18 ± 0.10 a
B14	3.93 ± 0.23 a	2.63 ± 0.11 a	2.06 ± 0.06 bc
B15	3.42 ± 0.23 bcd	2.49 ± 0.29 ab	1.86 ± 0.17 d
B16	3.49 ± 0.31 bc	2.56 ± 0.12 ab	1.98 ± 0.11 cd
BY1	3.35 ± 0.25 bcd	1.90 ± 0.11 de	2.16 ± 0.08 a
BY3	3.44 ± 0.31 bc	2.16 ± 0.14 c	2.04 ± 0.13 bc
BY4	3.60 ± 0.22 b	1.95 ± 0.01 de	2.13 ± 0.05 b

Different alphabets within a column mean a significant difference among the strains.

**Table 3 foods-11-02952-t003:** Re-screening of proteases-producing strains.

Strains	Enzyme Activity (U/mL)			
	Neutral Protease	Alkaline Protease	Acid Proteinase	Total Protease
B16	22.62 ± 0.91 a	2.54 ± 0.51 a	0.46 ± 0.26 c	25.61 ± 1.68
BY4	22.27 ± 2.00 a	0.94 ± 0.16 def	1.60 ± 0.13 ab	24.81 ± 2.29
BY1	21.56 ± 0.17 a	2.62 ± 0.29 a	--	24.18 ± 0.46
BY3	21.97 ± 2.44 a	0.67 ± 0.23 ef	0.96 ± 0.29 bc	23.60 ± 2.96
B15	4.42 ± 0.53 b	1.37 ± 0.27 bcd	0.69 ± 0.66 c	6.48 ± 1.46
B14	3.45 ± 0.23 bc	2.01 ± 0.27 b	1.83 ± 0.88 a	7.29 ± 1.38
B1	2.69 ± 1.31 bcd	1.98 ± 0.55 b	--	4.67 ± 1.86
B13	2.44 ± 0.38 cde	0.80 ± 0.17 ef	0.73 ± 0.11 c	3.97 ± 0.66
B9	1.30 ± 0.13 de	1.16 ± 0.22 cde	--	2.46 ± 0.35
B2	1.21 ± 0.21 de	0.53 ± 0.20 f	1.14 ± 0.15 abc	2.88 ± 0.56
B10	0.84 ± 0.26 de	0.73 ± 0.19 ef	--	1.57 ± 0.45
B5	0.63 ± 0.15 e	1.57 ± 0.37 c	0.54 ± 0.08 c	2.74 ± 0.60

Different alphabets within a column mean a significant difference among the strains. “--” represents no protease activity detected.

## Data Availability

Data is contained within the article.

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
