# Peer review of "Enhancement of Polypeptide Yield Derived from Rapeseed Meal with Low-Intensity Alternating Magnetic Field"

_foods, 2022, doi:10.3390/foods11192952_

Round 1

Reviewer 1 Report

The introduction needs to be rewritten because the paragraphs seem like loose ideas that don't fit together very well. He should devote a couple of paragraphs to the plant and its cultivation. Then we could talk about the food by-product, its composition and the studies that exist on this by-product. He would continue with the biological activities and the obtaining of bioactive hydrolysates.

Authors must abide by the Foods journal guidelines if they decide to use their word template. The quotes are made in an incorrect way, they must be with numbers between brackets [1], [2-5]...

The paragraphs must be adjusted to the columns of the template respecting the margins, the font size and the font. The format of the references

The style of the bibliographic references is also not correct for the MDPI format.

It is difficult to read the article because there are sentences that are cut off and misspelled. Please review the document because there seems to be an error copying the document.

Figure 6B has three Y axes (polypeptides, CFU and pH), it must be indicated in another way because it seems confusing and looks like a cut figure.

It should be checked that the expressions are consistent throughout the manuscript, for example hours and h are used.

The proper use of the names of microorganisms should be reviewed. (Bacillus. Velezensis), B. Velezensis, B. velezensis and other forms are used. Establish a single criterion and make it homogeneous and correct.

The conclusions must be rewritten because they should not contain numerical data of the results. These data were already given in the abstract and in the results.

Best regards

Reviewer 2 Report

2.6.2. Optimization of magnetic field parameters

The authors must present a table where the variables and conditions under which the experiments were carried out to perform the Optimization are easily identified in the magnetic field.

2.8 statistical analysis

did not mention how they performed the statistical analysis for Optimization

Table 1 does not show if there is a statistically significant difference between the different batches

3.3. Optimization of cultural conditions

The Optimization is presented concerning each variable; however, the analysis does not allow reaching the optimal fermentation conditions with statistical validation. Consider upgrading it.

Round 2

Reviewer 1 Report

Thanks to the authors for accepting the changes in the manuscript. I have no more comments

Best regards